# Indications of Persistent Glycocalyx Damage in Convalescent COVID-19 Patients: A Prospective Multicenter Study and Hypothesis

**DOI:** 10.3390/v13112324

**Published:** 2021-11-21

**Authors:** Richard Vollenberg, Phil-Robin Tepasse, Kevin Ochs, Martin Floer, Markus Strauss, Florian Rennebaum, Iyad Kabar, Alexandros Rovas, Tobias Nowacki

**Affiliations:** 1Department of Medicine B for Gastroenterology, Hepatology, Endocrinology and Clinical Infectiology, University Hospital Muenster, 48149 Muenster, Germany; Kevin.Ochs@ukmuenster.de (K.O.); Florian.Rennebaum@ukmuenster.de (F.R.); iyad.kabar@ukmuenster.de (I.K.); tobias.nowacki@ukmuenster.de (T.N.); 2The First Department of Medicine, Klinikum Ibbenbüren, 49477 Ibbenbüren, Germany; martin.floer@gmail.com; 3Department of Cardiology I-Coronary and Peripheral Vascular Disease, Heart Failure Medicine, University Hospital Muenster, 48149 Muenster, Germany; Markus.Strauss@ukmuenster.de; 4Department of Medicine D, Division of General Internal and Emergency Medicine, Nephrology, and Rheumatology, University Hospital Muenster, 48149 Muenster, Germany; Alexandros.rovas@ukmuenster.de; 5Department of Medicine, Gastroenterology, Marienhospital Steinfurt, 48565 Steinfurt, Germany

**Keywords:** SARS-CoV-2, COVID-19, syndecan-1, glycocalyx, long-COVID-19

## Abstract

The COVID-19 pandemic is caused by the SARS CoV-2 virus and can lead to severe lung damage and hyperinflammation. In the context of COVID-19 infection, inflammation-induced degradation of the glycocalyx layer in endothelial cells has been demonstrated. Syndecan-1 (SDC-1) is an established parameter for measuring glycocalyx injury. This prospective, multicenter, observational, cross-sectional study analyzed SDC-1 levels in 24 convalescent patients that had been infected with SARS-CoV-2 with mild disease course without need of hospitalization. We included 13 age-matched healthy individuals and 10 age-matched hospitalized COVID-19 patients with acute mild disease course as controls. In convalescent COVID-19 patients, significantly elevated SDC-1 levels were detected after a median of 88 days after symptom onset compared to healthy controls, whereas no difference was found when compared to SDC-1 levels of hospitalized patients undergoing acute disease. This study is the first to demonstrate signs of endothelial damage in non-pre-diseased, convalescent COVID-19 patients after mild disease progression without hospitalization. The data are consistent with studies showing evidence of persistent endothelial damage after severe or critical disease progression. Further work to investigate endothelial damage in convalescent COVID-19 patients should follow.

## 1. Introduction

The novel coronavirus SARS-CoV-2 is infecting ever-increasing numbers of people around the globe. While the infection results in mild to moderate symptoms in most individuals, it triggers severe illness with high mortality in a subgroup of patients. In addition to acute respiratory distress syndrome (ARDS), a variety of other symptoms involving different organ systems (acute kidney injury (AKI), acute cardiac injury, coagulopathy, thromboembolic complications including stroke and pulmonary embolism, and circulatory shock) have been demonstrated in COVID-19 patients [1,2]. SARS CoV-2 has been shown to bind to angiotensin-converting enzyme-2 (ACE2) receptors and infiltrates endothelial cells via this mechanism [3,4]. This has led to the hypothesis that COVID-19-mediated symptoms are possibly caused by a dysregulation of the vasculature [5,6,7]. Endothelial dysfunction may cause vasoconstriction with resultant organ ischemia, inflammation-induced tissue edema, and a procoagulant effect. The inner surface of all vascular endothelial cells is covered by the glycocalyx. This consists of proteoglycans, glycosaminoglycan side chains, and sialoproteins [8,9,10]. The glycocalyx plays an important role in microvascular and endothelial function. In the context of COVID-19 infection, inflammation-induced degradation of the glycocalyx layer in endothelial cells has been demonstrated. The proteoglycan syndecan-1 (SDC-1) is an important core protein of the endothelial glycocalyx and an established marker of glycocalyx injury [11,12].

The aim of this study was to investigate glycocalyx damage in convalescent COVID-19 patients.

## 2. Methods

### 2.1. Study Subjects and Samples

Thirty SARS-CoV-2-infected (nasopharyngeal swab and test by polymerase-chain reaction) individuals with mild disease course (no inpatient treatment) who recovered from infection were seen in our outpatient clinic. Only healthy patients with no known pre-existing conditions and no existing regular medication were seen in the outpatient clinic for this study. Twenty-four of them were manually selected to match the age of the inpatients and healthy controls.

Serum samples from hospitalized COVID-19 patients (*n* = 54) with laboratory-confirmed SARS-CoV-2 infection (nasopharyngeal swab and test by polymerase-chain reaction) were collected at the Department of Gastroenterology, Hepatology, Endocrinology and Clinical Infectiology, University Hospital Muenster, Germany, and the Department of Gastroenterology, Marienhospital Steinfurt, Steinfurt, Germany (03/2020–03/2021). Samples were collected in the first 48 h after hospital admission. Disease severity was defined using the World Health Organization (WHO) severity categorizations of critical (requires life sustaining treatment, presence of acute respiratory distress syndrome (ARDS), sepsis, septic shock), severe (oxygen saturation <90% on room air, signs of pneumonia, signs of severe respiratory distress), or non-severe (absence of signs of severe or critical disease). ARDS was diagnosed according to the Berlin definition (bilateral opacities on chest radiograph, exclusion of other causes of respiratory failure) [13]. COVID-19 patients were categorized according to their respective worst conditions over the course of hospitalization. Of these patients, 18 had a mild disease course. Ten age-matched patients with mild disease course were selected and used as hospitalized controls.

Age-matched subjects without preexisting conditions were used as healthy controls (*n* = 13). Sampling was performed before the outbreak of the SARS-CoV-2 pandemic. The study was approved by the local ethical committees of the University Hospital Muenster and the University Hospital Goettingen (Approval No. 2020-210-s-S, 2020-220-f-S, and 25/8/14), and informed consent was obtained from each participant.

### 2.2. Quantification of Serum Markers

Plasma concentration of SDC-1 was measured by Diaclone CD138 ELISA (Diaclone Research, Besancon, France) according to the manufacturer’s instructions. SDC-1 levels were measured in ng/mL. Clinical laboratory assessments included complete blood count, creatinine, bilirubin, aspartate aminotransferase (AST), γ-glutamyltransferase (γ-GT), lactate dehydrogenase (LDH), creatine kinase (CK), C-reactive protein (CRP), albumin, ferritin, and interleukin-6, and were determined on the day of laboratory measurement and were used to characterize hospitalized patients’, outpatients’, and controls’ physiological conditions.

### 2.3. Statistical Analysis

For continuous variables, we report median with interquartile range. For categorical variables, we report absolute numbers and percentages. The Kruskal–Wallis test was conducted to compare groups. To compare subgroups, the Bonferroni correction post hoc test was performed when variance was equal (Levene’s test), and the Games–Howell test was performed when variance was different. For comparison of nominal scale level, group comparison was performed by chi-square four-field test. The Pearson correlation coefficient was determined to analyze the correlation of SDC-1 levels with clinical laboratory parameters. All tests were two-tailed and a *p*-value < 0.05 was considered to indicate a statistically significant difference. All statistical analyses were performed using SPSS 26 (IBM, Chicago, IL, USA).

## 3. Results

### 3.1. Cohort Characteristics

In the COVID-19 outpatient convalescent cohort, blood collection occurred at a median of 88 days after symptom onset (IQR 70–135 days). Blood sampling of hospitalized COVID-19 patients with non-severe course occurred a median of 6 days (IQR 2–17) after symptom onset. The convalescent COVID-19 patients, hospitalized patients, and healthy controls showed no significant differences regarding patient age and body mass index (BMI). There was no difference in gender between hospitalized COVID-19 patients and convalescent patients; significantly more women were included in healthy controls (*p* = 0.046). In contrast to the hospitalized patients, healthy controls and convalescent patients had no preexisting diseases. Hospitalized patients showed a mild disease course without need of oxygen supplementation (Table 1). Hospitalized patients with acute disease showed increased inflammatory laboratory parameters (ferritin, interleukin-6, and C-reactive protein) compared to convalescent patients and healthy controls (Table 2).

### 3.2. Significantly Increased Syndecan-1 Levels in Convalescent COVID-19 Patients

Convalescent COVID-19 patients (blood drawn at a median of 88 days after symptom onset) after a mild disease course showed significantly elevated SDC-1 levels compared to the healthy control population (*p* < 0.05) and to the hospitalized patients with acute disease (*p* < 0.01). SDC-1 levels of hospitalized patients were not significantly different from SDC-1 levels of convalescent patients (Figure 1).

### 3.3. Association of Laboratory Values with Syndecan-1

In the overall study group, there was a significant correlation between SDC-1 levels and laboratory parameters. Inflammatory parameters (LDH, *p* = 0.018; ferritin, *p* = 0.04, IL-6, *p* = 0.01, CRP, *p* = 0.04) correlated positively with SDC-1 levels, whereas albumin correlated negatively with SDC-1 parameters (*p* = 0.02) (Figure 2A–E).

## 4. Discussion

The glycocalyx plays a central role in endothelial and vascular regulation. The glycocalyx of the lung seems to be more susceptible to injury than that of the heart or brain [14]. Destruction of the glycocalyx exposes the endothelial cells to oxidative damage [12]. SDC-1 has an important function as a transmembrane receptor in the control of inflammation during influenza infection. Transmembrane SDC-1 has a regulatory impact on c-Met activity in influenza infections. Enhancement of c-Met activity results in anti-apoptotic signaling. This limits epithelial cell death following influenza infection [15]. In the context of severe infections, damage to the glycocalyx may be accompanied by increased concentrations of fragments in the blood. SDC-1 detaches from the surface of vascular endothelial cells, which thins and destabilizes the glycocalyx layer. This induces microvascular excessive permeability and extravascular leakage in microvessels and leads to interstitial edema [16]. In this study, we demonstrated that convalescent COVID-19 patients showed increased SDC-1 levels in a median of 88 days after disease compared to healthy controls as a sign of persistent glycocalyx damage after recovery from COVID-19.

Our study presented valuable results regarding cohort characteristics, which were in line with other studies. All study groups were age-matched. The percentage of males in the healthy control group was significantly lower than in the hospitalized patient group and the convalescent patient group, which may be a limitation of this study due to possible gender bias. COVID-19 inpatients and outpatients were comparable to previous publications in terms of age, comorbidities, and clinical presentation [11,17]. Hospitalized COVID-19 patients with acute disease showed significantly higher levels of inflammatory markers (CRP, IL-6) compared to convalescent patients and healthy controls as sign of inflammation due to acute viral infection [17]. 

Compared to healthy controls, hospitalized patients during acute disease showed significantly increased SDC-1 levels compared to healthy controls, but not when compared to convalescent patients. In the entire study cohort, correlation analysis provided evidence that higher inflammatory parameters (C-reactive protein, ferritin, interleukin-6) correlated significantly with elevated SDC-1 levels. These inflammatory parameters are established disease activity markers and predictors of worse outcomes in COVID-19 patients [18]. Elevated LDH levels [19], also an established marker of acute COVID-19, also correlated significantly with elevated SDC-1 levels. Our data are in line with data of Karampoor et al., who were able to demonstrate an increase in syndecan-1 levels in COVID-19 patients treated as inpatients, depending on the severity of the disease. Critically ill patients showed significantly increased syndecan-1 levels during the acute phase of the disease compared to moderately ill patients. It was shown that the determination of syndecan-1 on the day of admission is suitable for monitoring disease activity in addition to the determination of IL-6, IL-10, IL-18, and CRP [20] The glycocalyx consists of proteoglycans (e.g., SDC-1), glycosaminoglycans, and plasma proteins [21]. The glycocalyx represents an important regulator of endothelial cell homeostasis and immune response in the context of infection [9]. In the context of inflammatory conditions, this layer can be injured [22], and this can lead to tissue edema [23]. Our data support the hypothesis of glycocalyx damage during acute infection.

To our surprise, we observed significantly elevated SDC-1 levels in convalescent COVID-19 patients in a median of 88 days after symptom onset compared with healthy subjects without prior SARS-CoV-2 infection. This was demonstrated in patients after mild disease course, and without prior medical conditions and without regular medication. These results demonstrate for the first time elevated SDC-1 levels as an indication of persistent impairment of glycocalyx after COVID-19 infection with mild disease progression without hospitalization. These results support the work of Ambrosino et al. on hospitalized convalescent COVID-19 patients 2 months after severe or critical disease progression (WHO classification III/IV), in whom sex-dependent signs of endothelial damage were also found [24,25]. The study population, especially females, had significantly decreased endothelium-dependent flow-mediated dilation (FMD) compared with matched controls (demographics, previous disease). Finally, it has to mentioned that the small sample size in our study may be a relevant limitation.

The overrepresentation of women in the healthy cohort must be considered as a limitation of our study. Compared with the collective of healthy controls of Karampoor et al., the control collective used in this study showed comparable mean syndecan-1 levels (Karampoor et al. 24 (23–32) ng/mL versus 31.6 (17.1–54.7) ng/mL in our study) [20]. Endothelial damage correlated significantly with the severity of lung dysfunction (arterial oxygen tension, forced expiratory volume in 1 s, forced vital capacity, diffusing capacity for carbon monoxide). One cause of persistent endothelial damage after COVID-19 infection may be persistent immune activation. Chioh et al. recently demonstrated persistent immune activation and increased levels of circulating endothelial cells as signs of vascular injury in convalescent COVID-19 patients approximately 1 month after symptom onset [26]. This study included COVID-19 patients hospitalized in the setting of acute infection with varying degrees of disease severity.

## 5. Conclusions

Our study revealed elevated SDC-1 levels as an implication of persistent endothelial damage in non-pre-diseased, convalescent COVID-19 patients after mild disease progression without hospitalization. The data are consistent with studies showing evidence of persistent endothelial damage after severe or critical disease progression. Further prospective studies in convalescent COVID-19 patients depending on longer follow-up times and with larger numbers of subjects should follow to investigate persistent vascular damage.

## Figures and Tables

**Figure 1 viruses-13-02324-f001:**
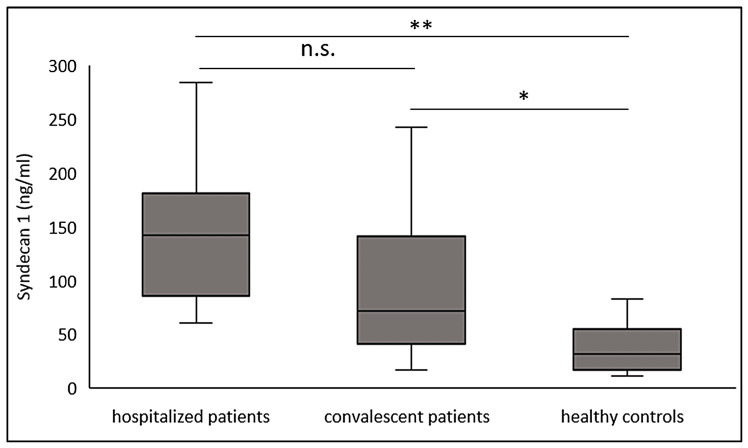
SDC-1 values of COVID-19 inpatient convalescent patients, hospitalized patients with acute disease, and healthy controls. * *p* < 0.05, ** *p* ≤ 0.01; n.s. not significant.

**Figure 2 viruses-13-02324-f002:**
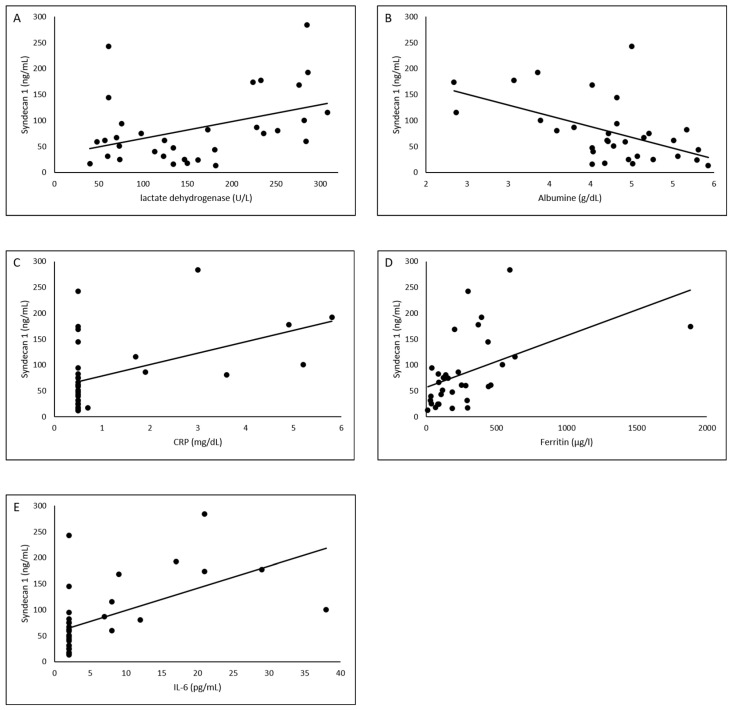
Significant correlations between SDC-1 levels and laboratory values in SARS CoV-2 patients, regression analysis plot. (**A**) lactate dehydrogenase, (**B**) albumine, (**C**) CRP, (**D**) ferritin, (**E**) Il-6.

**Table 1 viruses-13-02324-t001:** Cohort characteristics: differences were calculated by Kruskal–Wallis test; IQR, interquartile range; BMI, Body Mass Index.

Patients (Samples)		Hospitalized Patients (*n* = 10)	Convalescent Patients (*n* = 24)	Control Collective (*n* = 13)	*p*-Value
Patient Characteristics	Age, years median (IQR)	56.5 (54–63)	56 (55–58)	54 (51–57)	0.08
	Gender, male (%)	10 (100)	22 (91.6)	4 (31)	0.046
	BMI (IQR)	23.8 (22–25.9)	26.1 (22.8–28.1)	22.3 (20.2–25.7)	0.161
	Invasive Vent. (%)	0 (0)	0 (0)	0 (0)	1
	Oxygen therapy (%)	0 (0)	0 (0)	0 (0)	1
	Death (abs.)	0 (0)	0 (0)	0 (0)	1
	Interval 1. symptom to blood sample, days (IQR)	6 (2–17)	88 (70–135)		<0.001
SARS CoV-2 therapy	Remdesivir (5 days)	1 (10)	0 (0)	0 (0)	0.151
	Dexamethasone (6 mg, 10 days)	2 (20)	0 (0)	0 (0)	0.021
Pre-existing conditions	Chronic inflammatory disease (%)	4 (40)	0 (0)	0 (0)	<0.001
	Respiratory disease (%)	3 (30)	0 (0)	0 (0)	0.003
	Kidney insufficiency (%)	1 (10)	0 (0)	0 (0)	0.151
	Metastatic neoplasm (%)	1 (10)	0 (0)	0 (0)	0.151
	Diabetes (%)	1 (10)	0 (0)	0 (0)	0.151
	Arterial hypertension (%)	4 (40)	0 (0)	0 (0)	<0.001
	Coronary heart disease (%)	2 (20)	0 (0)	0 (0)	0.021
Medication	Angiotensin-1 receptor antagonist (%)	1 (10)	0 (0)	0 (0)	0.151
	Angiotensin converting enzyme inhibitor (%)	2 (10)	0 (0)	0 (0)	0.021

**Table 2 viruses-13-02324-t002:** Cohort characteristics: differences were calculated by Kruskal–Wallis test; IQR, interquartile range; LDH, lactate dehydrogenase; CRP, C-reactive protein.

	Hospitalized Patients *(n* = 10)	Convalescent Patients (*n* = 24)	Healthy Controls (*n* = 14)	*p*-Value
Syndecan-1 (ng/mL), median (IQR)	142 (85.2–181.3)	71.2 (40.1–141.5)	31.6 (17.1–54.7)	<0.001
Lymphocytes (rel., %), median (IQR)	22.2(16.5–33.2)	28.5 (27.2–34.4)	n.d.	0.23
Creatinine (mg/dL), median (IQR)	0.95 (0.88–1.3)	0.9 (0.8–01.0)	1 (0.8–1.2)	0.77
Bilirubin (mg/dL), median (IQR)	0.4 (0.3–0.7)	0.5 (0.4–1.0)	0.5 (0.2–0.7)	0.43
AST (U/L), median (IQR)	36 (26–52)	28 (25–34)	29 (22–35)	0.63
Gamma-GT (U/L), median (IQR)	82 (40–146)	27 (22–46)	22 (12–24)	0.003
LDH (U/L), median (IQR)	279 (232–285)	61 (57–74)	149 (127–179)	<0.001
CRP (mg/dL), median (IQR)	2.5 (0.5–5)	0.5 (0.5–0.5)	0.5 (0.5–0.5)	<0.001
Albumin (g/dL), median (IQR)	3.4 (2.7–3.9)	4.4 (4.3–4.5)	4.9 (4.1–5.3)	<0.001
Ferritin (µg/L), median (IQR)	383 (220–606)	252 (89–298)	87 (42–178)	0.002
Interleukin-6 (pg/mL), median (IQR)	14.5 (8–23)	2 (2–2)	2 (2–2)	<0.001

## Data Availability

Because of patient identifying data, original data are not available.

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
