# Peer review of "Indications of Persistent Glycocalyx Damage in Convalescent COVID-19 Patients: A Prospective Multicenter Study and Hypothesis"

_viruses, 2021, doi:10.3390/v13112324_

Round 1
Reviewer 1 Report
Summary: In the present manuscript, the authors measured the levels of proteoglycan syndecan-1 from the plasm of convalescent COVID-19 patients, hospitalized COVID-19 patients, and healthy controls to investigate the connection between glycocalyx health and COVID-19. They found that convalescent COVID-19 patients showed increased SDC-1 levels compared to healthy controls as a sign of persistent glycocalyx damage after recovery from COVID-19. Moreover, they observed the significant correlations between blood inflammatory markers and SDC-1 levels which led them to conclude that acute infection during COVID-19 cause glycocalyx structural damage. The methodology of this study are sloppy and a major, although the findings do fairly support the conclusion. This was an retrospective study, thus, it is expected that some control might be lacking, however, there are several major flaws with the methodology and data presented that are addressed in the comments below.
Major Comments:
1. The participants used in the study are a major concern. There is no description of how these groups were selected other than being ‘manually selected’. The groups were said to be matched, however, they are not remotely matched for sex. The hospitalized group is 100% men, while the controls are 70% women. Moreover, there are hardly any data on subject characteristics other than age and sex. Not even BMI is listed. There are no inclusion/exclusion criteria listed in the manuscript, which conveys that no criteria were used to include subject other than age. They claim that the controls were without pre-existing conditions, however, in table 1 it does not reflect that, as under the column for p-value it says n.d. (no data). They have 0 listed in the pre-existing conditions and medications section for convalescent plasma patients and controls. A statistical analysis can still be done if the values are 0. It is hard to believe that in 37 me and women in their 50s that none of them are on any medications or have any pre-existing conditions. For all of these reasons listed above there are major concerns on the quality of vetting that was done on participants that were included in this study.
2. The comparison between hospitalized patients 6 days after COVID symptoms vs. convalescent plasma patients 88 days after COVID symptoms is not likely to be a useful comparison. The findings could be interpreted to say that convalescent plasma treatment leads to long-term glycocalyx degradation, however, there are no long-term data on COVID patients that do not receive convalescent plasma. Perhaps, the comparison between the healthy control group would be more appropriate than a COVID control group that is not time matched. It is unfortunate that the healthy control group is so poorly matched with the other patients in this study.
Reviewer 2 Report
The authors evaluated SDC-1 in plasma samples of convalescent non-hospitalized patients with COVID-19, hospitalized patients with mild COVID-19 and control non-infected patients, observing elevated SDC-1 levels in both hospitalized and convalescent patients compared to control ones, thus supporting undergoing endothelial damage. Some points need to be clarified:
- groups of patients need to be better described in terms of severity (refer to WHO grading scale) and potential selection biases in the group of non-hospitalized patients should be eventually discussed or, otherwise, it is advisable to state that no potential biases have been identified
- time of sampling of hospitalized patients (as time from symptoms onset) need to be reported: during hospitalization, at the beginning of the diseases or at the end of clinical course.
- if available, I would suggest to add D-dimer among biochemical markers
Overall, a comparison with severe patients and the absence of longitudinal sampling would have added much information regarding the potential clinical application of SDC-1 dosing in clinical practice, in terms of anticipating clinical deterioration or monitoring the course of the diseases. I would suggest to discuss these issues
Reviewer 3 Report
By a prospective multicenter study, Vollenberg et al. demonstrated the persistent endothelial glycocalyx damage in convalescent COVID-19 patients. This prospective, multicenter, observational, cross-sectional study analyzed SDC-1 levels in 24 convalescent patients that had been infected with SARS-CoV-2 with mild disease course without the need of hospitalization. Authors claimed that this study is the first to demonstrate signs of endothelial damage in non-pre-diseased, convalescent COVID-19 patients after mild disease progression without hospitalization. The authors used ELISA for the measurement of the plasma concentration of SDC-1.
I have a few comments:
- The Sample size is the major limitation of this study to predict the correlation between SDC-1 and COVID-19-induced endothelial damage.
- Did the authors examine the plasma levels of D-dimer and TAT? It would be interesting to see the regression analysis plots comparing D-dimer and TAT with SDC-1.
